# α-Synuclein Strains and Their Relevance to Parkinson’s Disease, Multiple System Atrophy, and Dementia with Lewy Bodies

**DOI:** 10.3390/ijms241512134

**Published:** 2023-07-28

**Authors:** Noah J. Graves, Yann Gambin, Emma Sierecki

**Affiliations:** EMBL Australia Node for Single Molecule Sciences and School of Biomedical Sciences, Faculty of Medicine, The University of New South Wales, Sydney, NSW 2052, Australia; n.graves@unsw.edu.au (N.J.G.);

**Keywords:** α-synuclein, amyloid fibrils, Parkinson’s disease, dementia with Lewy bodies, multiple system atrophy, strain

## Abstract

Like many neurodegenerative diseases, Parkinson’s disease (PD) is characterized by the formation of proteinaceous aggregates in brain cells. In PD, those proteinaceous aggregates are formed by the α-synuclein (αSyn) and are considered the trademark of this neurodegenerative disease. In addition to PD, αSyn pathological aggregation is also detected in atypical Parkinsonism, including Dementia with Lewy Bodies (DLB), Multiple System Atrophy (MSA), as well as neurodegeneration with brain iron accumulation, some cases of traumatic brain injuries, and variants of Alzheimer’s disease. Collectively, these (and other) disorders are referred to as synucleinopathies, highlighting the relation between disease type and protein misfolding/aggregation. Despite these pathological relationships, however, synucleinopathies cover a wide range of pathologies, present with a multiplicity of symptoms, and arise from dysfunctions in different neuroanatomical regions and cell populations. Strikingly, αSyn deposition occurs in different types of cells, with oligodendrocytes being mainly affected in MSA, while aggregates are found in neurons in PD. If multiple factors contribute to the development of a pathology, especially in the cases of slow-developing neurodegenerative disorders, the common presence of αSyn aggregation, as both a marker and potential driver of disease, is puzzling. In this review, we will focus on comparing PD, DLB, and MSA, from symptomatology to molecular description, highlighting the role and contribution of αSyn aggregates in each disorder. We will particularly present recent evidence for the involvement of conformational strains of αSyn aggregates and discuss the reciprocal relationship between αSyn strains and the cellular milieu. Moreover, we will highlight the need for effective methodologies for the strainotyping of aggregates to ameliorate diagnosing capabilities and therapeutic treatments.

## 1. Introduction

### 1.1. Parkinson’s Disease, Multiple System Atrophy, and Dementia with Lewy Bodies

Parkinson’s disease (PD), Dementia with Lewy Bodies (DLB) and Multiple System Atrophy (MSA) are adult-onset neurodegenerative disorders that result in the progressive loss of motor and/or cognitive functions and are linked to neuronal degradation, accompanied by deposition of proteinaceous aggregates, including α-synuclein (αSyn). MSA, DLB and PD are frequently diagnosed with the apparition of motor symptoms [1].

PD patients develop the typical motor symptoms associated with Parkinsonism (resting tremor, rigidity, bradykinesia, freezing of gait) and present a range of non-motor symptoms such as constipation, depression, or sleep disorder. The disease can, in some cases, advance to include mild cognitive impairment, psychosis, or dementia at the late stage. In DLB, on the contrary, variations in cognitive function and fluctuations are a core diagnostic feature [2]. DLB patients present with a wide range of cognitive, neuropsychiatric, sleep, motor, and autonomic impairments [2]. Visuoperceptual functions are disproportionately affected, and visual hallucinations are an early symptom of DLB. MSA patients can exhibit parkinsonism (MSA-P) or cerebellar ataxia (MSA-C). MSA is accompanied by severe autonomic dysfunction (e.g., urinary incontinence, orthostatic hypotension) and pyramidal symptoms.

A key differentiation between synucleinopathies [3] is the rate of disease progression. Life expectancy is slightly reduced in PD patients compared to the general population, with a worse prognosis for patients presenting mild cognitive impairment. By contrast, median survival after diagnosis of MSA or DLB is short; 6–9 years and 5–8 years, respectively [4]. Disease progression is also faster, with MSA patients suffering earlier on from severe generalized autonomic failure and accelerated loss of ambulation (MSA-P). In DLB, cognitive symptoms develop before or shortly after the motor symptoms, whereas these symptoms arise years after diagnosis in PD cases [4].

Further distinguishing features between diseases include variable responses to treatment. For example, MSA-P patients respond less to levodopa treatment than PD patients. Only 30% of MSA-P patients see improvements with levodopa treatment, and the observed effects are often modest. Deep brain stimulation is also ineffective in MSA [1,3]. The motor symptoms of DLB patients can be relieved with levodopa, but treatment may worsen the neuropsychiatric symptoms. This is also true for other PD medications, such as dopamine agonists or MAO-B (monoamine oxidase B) inhibitors that may induce psychosis [5].

The striking differences in the development of the diseases arise from targeting different brain regions [6]. Indeed, the presence of αSyn inclusions and cell death appear with different spatial patterns in the different synucleinopathies. Aggregated αSyn is found mainly in the midbrain of PD patients, more precisely in the substantia nigra pars compacta, with the limbic and cortical regions potentially becoming affected at advanced stages of disease progression. In DLB patients, the cerebral cortex is affected early, as well as the limbic system and hippocampus. In MSA patients αSyn aggregation is detected in the olivopontocerebellar, nigrostriatal and autonomic systems. αSyn inclusions are also found in different cell types, with Lewy bodies (LBs) and Lewy neurites (LNs) existing mostly in the cytoplasm and axons of neurons in PD and DLB, and glial cytoplasmic inclusions (GCIs) or Papp-Lantos bodies present in the cytoplasm of oligodendrocytes in MSA [7]. LBs and GCIs have different morphologies and probably different compositions; LBs are rounder and more compact than GCIs in general [8]. Nevertheless, αSyn is the main constituent of both LBs and GCIs in PD, DLB, and MSA, respectively. Given the clear divergences in clinical and pathological phenotypes between synucleinopathies, one may wonder how and why the aggregation of αSyn differs across pathologies and, therefore, uniquely affects different cell types. Before reviewing these differences between aggregates and investigating whether intrinsic or extrinsic factors are at play, let us briefly introduce the underlined protein and its ability to aggregate.

### 1.2. Alpha-Synuclein: From Abundant Neuronal Protein to Major Neurodegenerative Culprit

αSyn is a small 140 amino-acid protein expressed throughout the brain and peripheral nerves as well as in other tissues at varying levels [9]. As shown in Figure 1, αSyn is composed of (1) a N-terminal amphipathic helix (aa 1–60) that is thought to mediate αSyn interactions with a lipid membrane and hypothesized to dictate fibril formation [10], (2) a central core region (aa 61–95) known as the non-amyloid-β component (NAC) that drives aggregation and amyloid formation, and (3) an acidic, proline-rich C-terminal tail (aa 96–140) that is negatively charged, predominantly remains disordered, and is thought to mediate protein-protein interactions [9,11]. The truncation of said carboxy-terminal regions results in increased filament formation [9]. Additionally, the N-terminal region has been shown to possess familial mutations [12] and, more recently, established a juvenile-onset synucleinopathy insertion in the same region [13].

Physiologically, full-length αSyn exists mainly as an unfolded monomer and is often seen as a model of intrinsically disordered protein. Its function has been described elsewhere [14]. Its most studied role is as a pre-synaptic protein that can regulate exocytosis and neurotransmitter release directly (by interacting with vesicles) and indirectly (as a molecular chaperone to the SNARE complex). Most of the focus on αSyn is linked to its role in synucleinopathies as, despite its abundance (1% of cytosolic proteins), knock-out of αSyn only leads to mild phenotypes [15]. Further links between αSyn and PD were established when missense mutations and later duplications and triplications in the SNCA gene were identified in cases of hereditary PD [9,16]. Despite these links, the involvement of αSyn in PD progression remained indeterminate due to the lack of a convincing association between αSyn deposition and neuronal death [17]. The discovery that αSyn aggregates could propagate between individual cells to different regions of the brain in a prion-like manner strengthened the case for a causal link between αSyn aggregation and PD progression [17]. Additionally, smaller oligomers of αSyn were found to be toxic to the cells, further involving αSyn in mediating neuronal death [11]. Finally, the development in recent years of amplification assays to detect the presence of αSyn aggregates in brains and biofluids proved that αSyn aggregation could be used as a long-awaited biomarker for the detection of PD and other syneucleinopathies [18,19]. Progress in the understanding of the pathological role of αSyn aggregation has been paralleled and supported by a strong biochemical and biophysical characterization of the aggregation process [11,20,21]. As stated, αSyn mainly exists as an unfolded monomer both in the cell and in vitro. When incubated at high concentration (high μM-low mM) in the presence of interfaces (e.g., shaking or presence of beads), αSyn first forms a variety of oligomers that slowly transform to contain parallel β-sheet structures [11]. These oligomers become elongation-competent and can grow by recruiting and converting monomers at an accelerated rate, forming amyloid fibrils with a distinctive cross-β structure [11].

### 1.3. Synuclein Aggregation as a Diagnostic Biomarker

The discovery of biomarkers that can be used to establish a diagnosis in the early progression of a disease is undoubtebly valuable for most pathologies but becomes absolutely critical for presently irreversible neurodegenerative disorders. In vitro, amplification of αSyn aggregates by real-time quaking-induced conversion (RT-QuIC), or protein misfolding cyclic amplification (PMCA), has provided a breakthrough in the long-awaited ability to detect PD [19,22]. Using either technique, different groups have been able to differentiate PD patients from healthy controls, but also Alzheimer’s disease patients, by analyzing the seeding propensity of their biofluids [23,24]. Macroscopically, αSyn aggregation, as visualized by the fluorescence of the dye Thioflavin T, presents a lag period before fibrillating in an exponential fashion. The lag time is dramatically shortened by the introduction of previously formed nuclei, a property that diagnostic seed amplification assays (SAA) take advantage of [19]. The principle of the SAA is described in Figure 2. These assays allow for the potential to differentiate people with synucleinopathies earlier from healthy controls.

In a very recent study from Siderowf et al., the well-characterized multicenter Parkinson’s Progression Markers Initiative (PPMI) cohort was used to further assess the diagnostic performance and capabilities of αSyn SAA [25]. This study represented the largest single analysis for αSyn SAA in the history of the field and provided a broad understanding of both the sensitivity and specificity of SAA in Parkinsonism diagnosis. Over the 9 years the analysis was conducted, 1123 participants were studied [25]. Of these patients, 545 were positive for PD, 163 were healthy controls, 54 were participants with scans without evidence of dopaminergic deficit, 51 were prodromal participants, and 310 were non-manifesting carriers. Sensitivity for PD detection in these assays was found to be 87.7%, while specificity for healthy controls was 96.3%. The sensitivity of the αSyn SAA in sporadic PD with the typical olfactory deficit was 98.6%, comparatively [25]. The proportion of positive αSyn SAA was lower in subgroups, including Leucine-rich repeat kinase 2 (LRRK2) PD at 67.5% positivity and participants with sporadic PD without olfactory deficit at 78.3% positivity [25]. Participants with the LRRK2 variant and normal olfaction had an even lower αSyn SAA positivity rate sitting at only 34.7% detection positivity. Among prodromal and at-risk groups, 44 of 51 (86%) of participants with idiopathic Rapid eye movement sleep behavior disorder (RBD) or hyposmia had positive αSyn SAA (16 of 18 with hyposmia, and 28 of 33 with RBD) [25]. Although this study provides a detailed look into the current practices of SAA, several limitations restrict current seed amplification assays from being the be-all-end-all synucleinopathy diagnostic tool. Among sample size limitations and difficulty in establishing accurate results with varying genetic forms of PD [25], the main determining limitation is that the assay is not yet sufficiently quantitative. To perform a SAA, the original aggregated seed must undergo several rounds of amplification to produce a conclusive positive or negative result. With those multiple rounds of amplification, it is difficult to assess the original number of aggregates, which removes the capability to monitor active disease progression throughout the patient’s diagnosis. Longitudinal research remains key in investigating the prognostic value of αSyn SAA and whether changes in quantitative measures of αSyn aggregation could indicate progressive pathology over time [25].

Iranzo and colleagues [26] applied RT-QuIC to samples from patients with idiopathic rapid eye movement sleep behavior disorder (iRBD) at a possible prodromal stage of synucleinopathies. Disease-positive outcomes were partially related to the follow-up diagnosis of PD or DLB, validating the method for early diagnosis [26,27]. Systematic analysis of two experimental cohorts showed promising results in the identification of probable DLB in patients with mild cognitive impairment [28]. So far, cerebrospinal fluid has proven to be the best medium for both sensitivity and selectivity, but ongoing investigations show that more accessible samples [29] (including olfactory mucosa [27,30] or skin biopsies [31]) can be used.

Serial propagation or the ability to imprint a conformational strain on newly formed fibrils can be exploited to amplify pathological αSyn aggregates [32]. Indeed, multiple rounds of in vitro amplification, using monomeric recombinant αSyn as substrate and brain homogenates or CSF as templates, successfully produced recombinant fibrils “imprinted” by the pathogenic strains [32]. Multiple generations of fibrils showed similar structural profiles based on secondary structure assessment and protease resistance, although the extent of the structural fidelity at the atomic level is still unclear [32].

Amyloid fibrils are a common fold in nature [33]. Importantly, other proteins involved in neurodegeneration, such as amyloid β, Tau, and the prion PrPC have been shown to form such fibrils [34]. Studies of prion diseases have established that the amyloid fibrils of PrPSc, containing multiple filaments, do, in fact, exist in different conformations. These conformations or strains possess different structural, biophysical, and biochemical properties and lead to markedly different pathologies. Likewise, the concept of different strains mediating dissimilar pathologies is emerging in tauopathies [35], Alzheimer’s disease [36] and, as reviewed here, synucleinopathies.

## 2. Distinct αSyn Strains Lie at the Core of PD, MSA & DLB

### 2.1. MSA and PD αSyn Fibrils Are Structurally Different

As introduced before, αSyn aggregates present in PD, DLB, and MSA appear in different cellular contexts (glial cells or neurons) and form different types of inclusions (GCIs or LBs) [16]. Both GCIs and LBs are composed of fibrillated αSyn, often highly phosphorylated, acetylated and variously truncated. They are both reactive to ubiquitin and were found to contain lipids, organelles and a variety of proteins. Their gross morphologies are different, with LBs appearing as dense, round inclusions (brainstem LBs in particular), while GCIs appear as triangular, sickle, or conical forms, suggesting compositional and organizational differences. This is also demonstrated by the distinct response to staining where GCIs are reactive to both the Campbell–Switzer and Gallyas-Braak methods, while LBs are only poorly visualized using the latter [37]. Analysis of the secondary structure of the inclusions in tissues also pointed to a difference in the β-sheet content of the aggregates, with LBs presenting a higher proportion of β-sheet structure compared to GCIs [38]. Such a difference could arise from a different composition of the inclusions (i.e., a variable percentage of αSyn) or structural differences in the fibrils. Furthermore, multiple groups have shown that αSyn filaments isolated from LBs and GCIs have distinct responses to proteolysis and reactivity to conformational antibodies.

Recently, Schweighauser et al. obtained the structures of αSyn filaments isolated from the brains of MSA patients [39]. Cryo-EM revealed two types of filaments, both composed of two protofilaments, with atomic resolution (2.6–3.1 Å). In both cases, asymmetrical packing of the two protofilaments and the presence of a central cavity incorporating non-proteinaceous molecules were noted [39]. These conformations depart significantly from previously solved structures of αSyn fibrils that had been generated in vitro. Tellingly, attempts by the authors to solve the structure of αSyn filaments isolated from DLB patients’ brains initially failed due to the lack of features. Indeed, DLB filaments were found to be thinner and did not appear to twist, further suggesting the existence of different conformations or strains between DLB and MSA. This was proven 2 years later when Yang et al. managed to solve the structure of a sub-population of twisted DLB filaments [9]. The structure obtained in this case was indeed dramatically different from the previously solved structures. Strikingly, the “Lewy fold” observed in DLB and PD in this study only contains one protofilament compared to two in the MSA folds, as outlined in Figure 3. The presence of an extra-density coordinated by four lysines is still present, but the amino acids involved are different. The authors also noted the presence of peptidic “islands” that pack against the fibrils’ core [9]. These densities were not observed in the MSA folds.

The difference in twisting has also been observed on fibrils amplified from patients. Using recombinant fibrils obtained by amplification of seeds from cerebrospinal fluid (CSF) of PD and MSA patients, Shahnawaz et al. [40] showed different conformations, noting in particular that MSA filaments had shorter twists (65.2 nm average) compared to PD filaments (108.5 nm). Other groups have since studied the structures of the αSyn filaments post-amplification [41,42,43]. These studies revealed a variety of folds. Whether this is reflective of true heterogeneity between patients or experimental bias will require further investigation. It has been reported that amplification does not always replicate the properties of the seeds. The variation could be linked to different amplification protocols. Alternatively, this may reflect changes in seed properties in different tissues (brain vs. cerebrospinal fluid) or at different clinical phases. Nevertheless, studies where samples amplified from MSA and PD patients were simultaneously observed validated the view that PD and MSA αSyn fibrils were structurally different. Frieg et al. [41] noted that although the structure of the individual protofilaments was similar, the quaternary arrangements were dissimilar, as was also shown by Burger et al. [43]. Note, however, that the resulting structures from those studies did not overlap.

These recent discoveries prove that structural differences exist between αSyn filaments linked to different pathologies, reinforcing the idea that αSyn aggregation is a central feature in the pathogenesis of synucleinopathies. This could suggest that heterogeneity exists within each disease group and maybe even in each patient.

### 2.2. PD, MSA and DLB-Derived Fibrils Induce Different Pathologies

The observed variety of conformations between synucleinopathies is paralleled with varying pathological properties. The concept that αSyn aggregation can propagate between cells and different regions of the brain through neuroanatomical connections—in a prion-like manner—has gained increasing support over the years [17]. Contrary to the true prion, though, propagation of αSyn pathologies between individuals remains controversial [17]. Nevertheless, it is now accepted that injection of αSyn aggregates from biological or synthetic origins can lead to the development of symptoms and neurodegeneration in transgenic, non-transgenic mice, and non-human primates [44].

Work by the group of Prusiner, Nobel Prize laureate for the discovery of prions, showed that injection of human MSA brain homogenates to transgenic mice expressing the A53T mutant human αSyn led to neurodegeneration accompanied by deposition of αSyn in neurons [44]. They also demonstrated serial propagation of infectivity, a critical trait of prions [45]. This led to the proposition that MSA should be treated as a prion disease [44]. Notably, these studies found that PD brain homogenates failed to produce a phenotype, suggesting the involvement of two separate strains. Further studies also showed that MSA aggregates are more potent than aggregates isolated from PD and DLB patients to induce neurodegeneration [46]. Similarly, Peng et al. [46] found that GCI-purified αSyn was a more potent seed compared to LB-αSyn, in vitro and in vivo. They also showed that neither strain was cell-specific. Recent work from the Goedert laboratory further refined this observation [47]. Using ultracentrifugation on brain homogenates, the authors showed that only fractions enriched in Sarkosyl-insoluble αSyn aggregates were seeding-competent, with MSA fractions being 100 times more potent than PD homogenates to induce aggregation in a cell model. It should be noted, though, that in these studies, MSA-related αSyn fibrils injection did not faithfully recapitulate the MSA pathology, affecting primarily neurons [48]. This is not true for non-human primates, where injections of GCI induce an MSA-like pathology with loss of oligodendrocytes and gliosis [49].

Brain-derived aggregates were also used as seeds for amplification, and the end products could be used for in vivo studies. These samples provided “cleaner” models for inoculation studies, as external factors are diluted, and only the conformation of the assembly should be maintained. Using such tools, Van der Perren et al. [50] obtained fibrils amplified from PD, MSA, and DLB patients and found a striking difference between MSA/PD fibrils and DLB fibrils (note that the finer structural details that distinguished MSA from PD fibrils were not obtained). DLB fibrils were unable to promote αSyn spreading or trigger neurodegeneration upon the nigral inoculation of rats and did not lead to the development of symptoms [50]. By contrast, both MSA and PD fibrils induced dopaminergic neuron degeneration, but only MSA fibrils led to spreading and immune response. This data reveals that at least some of the phenotypic traits of the different synucleinopathies are supported by the structural properties of different αSyn strains [50]. Interestingly, in a follow-up study by Peng et al., the group of Virginia Lee found that LB-amplified fibrils were more potent in inducing αSyn pathology than the original brain extracts in transgenic mice [51]. LBD-derived amplified fibrils also induced 5–50 times more neuronal inclusions than recombinant pre-formed fibrils (PFFs) and induced strikingly different pathologies [52].

Experimental results on the seeding propensity of different species are, at times, contradictory due to the variations in seed preparations and/or the selected models. Nevertheless, a consensus emerges [53] that MSA seeds are better suited to repeatedly induce pathology in animal models than PD and DLB seeds. Lastly, synthetic recombinant αSyn fibrils can induce aggregation in cells but do not induce pathology in mice models [54].

### 2.3. Structurally Different Recombinant αSyn Strains Lead to Distinct Pathologies

Additional intriguing observations support the idea that PD, DLB and MSA can be induced by different αSyn strains. The production of synthetic αSyn PFFs from recombinant monomeric αSyn generates a multiplicity of polymorphs upon variation of experimental conditions [11] (see Table 1).

Biophysicists identified multiple structural arrangements in not only fibrils but also oligomers. These αSyn oligomers exist as various assemblies (annular, round, tubular, etc.), differentially enriched in secondary structure (α-helix or β-sheet). External factors, such as small molecules or cations, were found to strongly impact the polymorphism of said αSyn oligomers. Not all structural variants are able to convert into fibrils, and importantly, they exert variable toxicity when applied to cells. Indeed, Danzer et al. [59] showed, for example, that annular, but not spherical or globular oligomers, were able to trigger Ca^2+^ influx and cell death in vitro. Conversely, only spherical and globular oligomers seeded αSyn aggregation in cells. Although oligomers may initially seem responsible for the concentrated toxicity of αSyn, it was mainly the fibrils that induced phenotypic responses. Recasens et al. [60] could induce pathology in wild-type mice upon cerebral inoculation with LB-derived insoluble fibrils but not using non-LB fractions that contain soluble oligomers. This must be nuanced, though by a recent study by Bourdenx et al. [61]; the authors showed that small soluble αSyn aggregates did not induce pathology in mice as expected, but in non-human primates, both LB-derived fibrils and non-LB oligomers had an effect. Nevertheless, Thomsen et al. demonstrated that striatal injections of αSyn PFF induced progressive pathological synaptic dysfunction prior to cell death that can be detected in vivo via PET (positron emission tomography) [62]. Upon intrastriatal injection of αSyn PFFs, a progressive αSyn pathology was observed from the PFFs, rather than the monomeric αSyn [62], with loss of dopaminergic and synaptic function accompanied by neuroinflammation. This was also the observation of Luk et al. [63] and many follow-up studies.

Recently, several groups have started investigating the effect of different polymorphs on pathology development. Peelaerts et al. [54] used structurally characterized polymorphs of αSyn aggregates, termed “ribbons” and “fibrils” [64] and investigated their effect after intracerebral inoculation of rats. Structurally, ribbons appear relatively flat and twisted compared to fibrils that contain no twists and are cylindrical in form [50]. Although ribbons produced LB/LN-like inclusions more efficiently, it was found that the fibrils led to increased cellular death in the striatonigral pathway. Suzuki et al. [56] also injected these two synthetic fibrils into WT mice and observed increased endogenous αSyn phosphorylation and ubiquitination with the low salt preparation (obtained in the same conditions as the ribbon strain). To apply this experiment to an MSA model, the same strains were injected into transgenic mice expressing αSyn in oligodendrocytes [65]. Again, the two strains behaved differently, with fibrils promoting high toxicity with severe myelin loss and neurodegeneration, while ribbons were less toxic but induced more αSyn deposition in astrocytes. This study has also been extended to other polymorphs. In Rey et al. [57], ribbons and fibrils, along with two types of aggregates, were injected into the olfactory bulb of mice. The observed data recapitulate in part the previous observations that ribbons spread more efficiently than fibrils, although another polymorph (called F-91) was the most efficient. Importantly, they also showed a variable ability to propagate along neuronal routes and cross synaptic relays. Fayard et al. [66] performed a similar experiment in non-human primates. Intra-putaminal injection of either ribbons or fibrils led to strain-specific patterns of αSyn-induced pathology measured by endogenous aggregation, phosphorylation at S129 and amplification propensity. In parallel, Lau et al. [58] used recombinant fibrils of the A53T mutant of αSyn formed upon incubation in high salt (“S fibrils”) or no NaCl (“NS fibrils”) that presented different biochemical profiles. Upon intracerebral injection in transgenic mice expressing human A53T αSyn, S fibrils led to a reduced incubation period before the development of motor symptoms [58]. The clinical signs also differed with S fibrils causing hind-limb paralysis and bradykinesia and NS fibrils leading to a shaking phenotype. Phospho-αSyn deposits were observed in varying brain regions, with the NS-injected fibrils exclusively developing inclusions in the olfactory bulb and hippocampus. Notably, those strains affected different cell types and led to different morphologies of the inclusions. S fibrils produced ring-like deposits reminiscent of neuronal cytoplasmic inclusions, specifically in neurons, whereas NS fibrils drove the formation of LB-like deposits in not only neurons but astrocytes as well [58]. Interestingly, the authors concurrently treated mice with brain homogenates from MSA and DLB patients or spontaneously ill homozygote mice (M83^+/+^). The phenotypes and inclusion morphologies induced by MSA-derived strains resembled those produced by S fibrils, while symptoms of NS-fibrils mirrored those generated by M83^+/+^-brain injections. MSA-derived aggregates and S fibrils were both found to be less conformationally stable than their counterparts (M83^+/+^ and DLB-derived fibrils and NS fibrils, respectively), suggesting an inverse relationship between conformational stability and strain propagation. Note that further structural details were not obtained [58].

Liu et al. [55] also generated recombinant αSyn fibrils formed under different experimental conditions and investigated their propagation propensity. Fibrils were formed under high and low salt conditions at neutral pH and in low pH conditions. Reinforcing the results from previous studies, αSyn PFFs formed at a lower salt concentration induced proficient αSyn aggregation compared to the high salt strain, as measured by phosphorylation of αSyn at the injection site. Further, αSyn aggregation propagated into vast brain regions upon inoculation with the low salt PFFs compared to high salt PFFs. This parallels the previous observations with ribbons, formed at low ionic strength, and fibrils, obtained at physiologically salted conditions [64]. Interestingly, αSyn aggregates obtained at low pH produced the strongest phenotype, including increased seeding, propagation ability, and amplified induction of motor neuron degeneration and cerebral inflammation. Strikingly, in low pH conditions, αSyn primarily forms amorphous aggregates, raising an important question concerning structural imprinting in this case.

Long et al. [67] recently validated that structural imprinting does indeed support the pathological potential of αSyn aggregates. In a recent study, they used a E46K αSyn mutant associated with familial, early-onset PD [68] to examine the link between structure and pathology. They generated PFFs from E46K αSyn, WT αSyn and PFFs termed hWTcs (for human WT cross-seeded), obtained from WT αSyn seeded by E46K fibrils. First, the structures of the fibrils were assessed by EM; both E46K and hWTcs PFFs presented a distinctive right-hand twist, while WT fibrils bore left-handed twists. Higher-resolution structural characterization by cryo-EM validated that hWTcs fibrils closely resembled the E46K fibrils and not the WT assemblies. When these PFFs were inoculated to mice, the pathology induced by hWTcs was also closer to the one arising from E46K PFF inoculation, with impaired motor function and exacerbated αSyn aggregation compared to WT fibrils. These results led the authors to conclude that hWTcs had “inherited” the pathological properties of the seeding E46K strain rather than the intrinsic properties of WT αSyn [68].

Taken together, these observations suggest that MSA and PD produce different αSyn strains that are “imprinting” the pathology while conjunctively revealing that the structural arrangements of αSyn fibrils encode enough information to lead to the development of distinct pathologies, as seen in Figure 4. Therefore, evidence arises for a strong correlation between pathologies and conformational strains, as in the case of the prion. This then raises the question of which comes first: the pathology creating the strain or inversely?

## 3. Strains and Pathology: The Role of the Environment

### 3.1. The Strain-to-Cell Connection

As described, pathological αSyn strains only affect certain types of cells. In most studies, αSyn deposition in neurons vs. astrocytes, for example, clearly delineates the ability of various strains to induce pathologies. Induced cellular death also differs and rarely correlates with the presence of inclusions. Selective response of cells might arise from various factors, including the cellular content in αSyn, the possibility of uptake, and induced toxicity through loss/gain-of-function.

Regardless of the strain, emerging evidence associates αSyn expression level and αSyn accumulation. Intracellular αSyn concentration affects the cell’s response to PFFs. Courte et al. treated neurons isolated from diverse parts of the brains with PFFs and found that the levels of phospho-αSyn accumulation were significantly different. In this case, the neurons from the hippocampus presented the most inclusions [69]. This was despite a similar intake with no observable effect on cell death. The amount of aggregation correlated with the endogenous expression level of αSyn when WT mice were used; reduction in αSyn deposition could be observed when neurons from heterozygous SNCA^+/−^ mice, with lower αSyn expression, were used. It was also observed that isogenic correction of αSyn levels in human induced pluripotent cells-derived dopaminergic neurons from SNCA triplication patients reduced intracellular αSyn aggregation and cell death after exogenous addition of PFFs or PD-/MSA-derived fibrils [70]. This is also in line with some observations of familial cases of synucleinopathies. SNCA copy number variations (CNVs), duplications [71], and triplications [72] have all been linked to familial forms of the disease. A clear dosage effect was reported, with the severity and rapidity of disease progression increasing with the copy number [73,74]. It is noteworthy that although CNVs primarily lead to LB pathology, duplications affect the brainstem region more frequently compared to triplication cases that tend to develop dementia, mimicking the differential spread of synucleinopathies. Yet, the substantia nigra that is mainly affected by PD is not the highest αSyn expressing region of the brain, and more strikingly, oligodendrocytes do not present high endogenous levels of αSyn, to the point where it was long thought that they did not express αSyn at all. Therefore, a higher endogenous concentration of αSyn in a cell may facilitate aggregation, but other factors are certainly at play.

Variable susceptibility between different cell types may also be partially explained by how efficiently αSyn seeds enter the target cells. αSyn aggregates have been shown to diffuse through the cellular membrane or be taken up by macropinocytosis [75,76] or phagocytosis in the case of microglia [77]. In addition to these non-specific mechanisms, αSyn is also actively transported by receptor-mediated cellular uptake [78], which can be cell-type specific [76,79]. An important finding by Mao et al. [80] was the identification of a membrane-receptor, lymphocyte-activation gene 3 (LAG3), with increased affinity for αSyn aggregates that mediates the transmission of the αSyn pathology [81]. However, LAG3 may only be one of the receptors involved in αSyn internalization [82]. Previously, heparan sulfate proteoglycans (HSPGs) [83], and particularly syndecan 3 [84], have been shown to bind and mediate the uptake of αSyn aggregates. Syndecans have also been suggested to play a role in mediating fibril formation of not only αSyn [85] but also Tau [84] and Aβ fibrils [86]. This internalization route seems important for oligodendrocyte uptake but is dispensable for microglial or astrocyte uptake, offering an example of selectivity. Interestingly, HSPGs mediate the uptake of a variety of misfolded protein aggregates [83], and specific uptake can be fine-tuned by varying length and sulfation patterns of the receptors [87]. Binding to and clustering of different membrane proteins by αSyn aggregates has been studied and shown to correlate with seeding propensity [88]. As immune cells, microglia and, to a lesser extent, astrocytes expose other classes of membrane receptors compared to neurons or oligodendrocytes. Toll-like receptors (TLRs), a class of pattern recognition receptors that probe for molecular patterns associated with pathogens or danger, can bind to or uptake αSyn aggregates. TLR2 [89] and TLR4 [90] bind to αSyn aggregates, triggering both their uptake and pro-inflammatory signalling [91]. Polymorphism creates differences in the exposed binding surfaces that target specific receptors. α3 unit of the Na^+^/K^+^ ATPase (α3-NKA) has been proposed to play a significant role in mediating the uptake and toxicity of αSyn aggregates [88]. Importantly, polymorphs have differential binding affinities for α3-NKA and other membrane proteins, providing a possible explanation for strain-specific cell susceptibility [92]. Besides the receptor- or endocytosis-mediated entry, αSyn fibrils can propagate between cells by other mechanisms, including direct membrane penetration or tunneling nanotubes [93]. Of particular interest is the presence of synuclein in extracellular vesicles (EVs). EVs are emerging key players in cell-to-cell communication in the brain and as biomarkers in neurodegenerative disorders [94]. So far, little is known about the link between strains and their presence in EVs; however, the compositions of EVs differentiate PD and atypical parkinsonism [95]. Further, the internalization and cellular processing of EVs appear to be cell-type dependent [95], which could partially explain the cellular selectivity of some disorders.

The modulation of interactions between αSyn aggregates and other proteins extends beyond cellular uptake and may contribute to differences in seeding and toxicity. Indeed, through binding to distinct partners, αSyn strains could differentially impact cellular processes, modulating the induced toxicity, as seen with membrane receptors. αSyn has been shown to interact with multiple partners due to its intrinsically disordered nature and aptitude to self-organize [96]. The structure of the fibril is directly linked to the selectivity of the interactome in that each strain uniquely exposes residues principally at the N- or C-termini, thus modulating the accessibility to protein-protein interactions. Stephens et al. recently determined that the more exposed the N-terminus and the beginning of the NAC region of αSyn are, the more aggregation-prone monomeric αSyn conformations become [97]. It was found that solvent exposure of the N-terminus occurs upon release of C-terminus interactions when calcium binds, but the level of exposure and αSyn’s aggregation propensity is sequence and post-translational modification dependent [97]. Landureau et al. [98] examined the solvent accessibility of the above-mentioned ribbons and fibrils [64], resulting from the in vitro aggregation of WT αSyn in different salt conditions. They found that the N-terminus of fibrils was exposed to solvent, but the same sequences were inaccessible in ribbons. This feature and differences in the C-terminal region probably contribute to the selective binding and inhibition of the proteasome by ribbon-like aggregates observed by Suzuki et al. [56].

αSyn aggregation has been found to be disrupting the autophagy pathway [99]. Physical interactions between αSyn and autophagy-related proteins from the GABARAP [100] (Gamma-aminobutyric acid receptor-associated protein) and LC3 [99] (Microtubule-associated protein 1A/1B-light chain 3) families only occurred with aggregated forms of αSyn. Of particular pathological relevance are interactions with molecular chaperones [101]. Heat-shock proteins (HSP) are charged with the ability to recognize, stabilize, refold, or tag misfolded proteins for degradation, therefore, play a prominent role in neurodegenerative diseases. αSyn aggregates have been found to bind to different members of the molecular chaperone family (including HSP90, HSC70, DNAJB6, CLU, CRYAB, BAG, etc.), impacting cellular uptake and aggregation kinetics. Post-translational modifications rely on the recognition and binding of kinases to their substrates; therefore, the detection of αSyn by FYN [102], GRK5 [103], or Abl [104] per se could substantially vary depending on conformations. A recent finding by Burmann et al. [105] presents a fascinating interplay between the two processes, showing that the phosphorylation pattern does influence chaperone selection.

We would be remiss to not briefly mention that αSyn is not the only protein to aggregate in PD, MSA, or DLB. Tau (Tubulin-associated unit) is a highly soluble protein and, like αSyn, is intrinsically disordered [106]. The microtubule-associated protein contributes to the stability of axonal microtubules in the brain and, in this role, is involved in the regulation of axonal outgrowth and transport [106]. The binding of Tau to microtubules is regulated by post-translational modification mainly via phosphorylation [106] and has been detected in some synucleinopathies. Both αSyn and Tau proteins are found in LBs, and the presence of neurofibrillary tangles is frequently noted [107]. The two disordered proteins have been found to co-aggregate [108], and αSyn fibrils can seed Tau aggregation [109]. Guo et al. [110] showed that repeated seeded fibrillation in vitro led to the creation of a new strain of αSyn, with an enhanced ability to induce αSyn and Tau aggregation in neurons, compared to the initial fibrils. Mixed αSyn/Tau fibrils have increased seeding activity in cells and in vivo [111]. This study further suggests that conformational strains of αSyn could partially explain the heterogeneity observed in the spectrum of synucleinopathies. Although Tau is not the main constituent protein in PD, MSA, or DLB as established, the emerging data on the presence of structurally different aggregates of Tau [35,110] is worth noting as it reiterates the importance in considering all aspects of synucleinopathy strainotyping.

### 3.2. Cell-to-Strain Imprinting

As described above, multiple factors can explain why specific strains target different cell types. Moving forward, we will explore how the reciprocal relationship in which the cellular environment can, in turn, construct conformational strains.

Intuitively, one would expect the cell environment to have a significant impact on the ab nihilo aggregation of αSyn. Peng et al. [46] showed that incubating monomeric αSyn in either oligodendrocytes or neuron lysates led to the creation of strains with different properties. In particular, oligodendrocytic strains led to increased development of a phospho-αSyn phenotype in vitro, similar to treatment with GCI-purified aggregates [46]. Identifying exactly which cellular factors affect strain selection is yet to be strictly determined as multiple parameters can influence αSyn conformation.

The cellular environment strongly influencing strain polymorphism is evidenced by the structures of the filaments themselves. As described above, αSyn aggregates isolated from GCIs are composed of two protofilaments arranged around a central cavity [39]. In this space, coordinated by multiple lysines, are unidentified non-proteinaceous molecules originating from the cell that would influence the packing of said protofilaments. Additional factors may also affect strain conformation, such as post-translational modifications and truncations, which have been shown to significantly modify aggregation kinetics and would presumably lead to the formation of various polymorphs. The previously mentioned study by Rey et al. [57] includes fibrils formed by C-terminally truncated αSyn, which have more spreading potential and a different distribution pattern compared to the corresponding fibrils made from full-length proteins. This evidence reinforces the notion that it would not be surprising that different patterns of post-translational modifications would generate unique strains. Similarly, molecular chaperones have also been shown to affect fibrillation, with different efficiencies depending on the kinetics of aggregation [112]. Polymorphs could be differentially repressed based on the prevailing chaperones leading to the selection of specific strains; however, it is still unknown at what stage of the aggregation process the conformational selection occurs.

Because αSyn is intrinsically disordered and folds upon binding to its partners, the selection of conformations is largely influenced by the environment [113], and it is possible that non-physiological interactions trigger conformational changes that put αSyn on the path of aggregation towards a specific strain. This seems to be the case with the interaction between αSyn and p25α, for example. p25α is a constituent of myelin, possesses a strong affinity for myelin basic protein (MBP), and is commonly located in oligodendrocytes. Notably, in MSA brain tissues, however, p25α re-localizes from the myelin to the cytosol [114]. p25α colocalizes with αSyn aggregates in cells and induces αSyn aggregation. In vitro, the aggregation of WT αSyn in the presence of sub-stochiometric amounts of p25α leads to the creation of structurally different fibrils compared to WT alone [115]. It was also determined that treatment of neurons with these αSyn/p25α fibrils led to the creation of larger cell inclusions, while intramuscular injection of the same PFFs reduced the life span of A53T αSyn transgenic (M83) mice and induced increased motor degenerative symptoms compared to WT PFFs [115].

In continuing with non-physiological interactions influencing αSyn aggregation, the cellular milieu has also recently been noted to play a key role in amplification potential. Although the serial propagation of a strain via in vitro amplified, fibrils achieved the same conformation as the seeds as previously noted [45,50], not all environments may be so permissive. It has been established that MSA strains propagate in cell lines expressing either WT or A53T human αSyn [53] but fail to induce aggregation in E46K αSyn-expressing cell lines. Other interesting observations by Peng et al. [46] suggest that the cellular milieu can either propagate or modify αSyn strains depending on cell type. First, synthetic αSyn PFFs, when passaged in cellular cultures of oligodendrocytes, neurons, or cortical neurons, acquired distinctive features depending on the cell type. Fibrils passaged in oligodendrocytes showed a superior ability to induce cellular αSyn aggregation, similar to GCI-isolated αSyn aggregates. Then, when the authors propagated LB-isolated αSyn aggregates in transgenic mice that only expressed αSyn in oligodendrocytes, the subsequently isolated aggregates behaved like MSA strains in terms of potency and spreading pattern upon intracerebral injection. The reverse, however, was not true and GCI-isolated strains retained their structural and functional characteristics upon passaging in neuronal cultures.

This idea can be taken one step further, and one could consider that the entry point of the αSyn aggregates, provided they did not arise spontaneously in the brain, could impact the ensuing pathology. The idea that PD, and potentially other synucleinopathies, originate in the periphery remains controversial. First formulated as the ‘dual hit’ hypothesis by Hawkes et al. [116], the assumption postulates that external factors such as infection or inflammation could induce the aggregation of αSyn in other parts of the body, in particular the olfactory bulb and the gut. This is supported by the presence of LBs in those regions in PD patients. The “gut-to-brain axis” hypothesis gained strong support from observations that severing the vagal nerve prevented the propagation of αSyn pathology in mice models treated with PFFs [117]. Vagotomy [118] and appendectomy [119] could potentially protect against PD development in retrospective cohort analysis, although results remain contentious [120,121]. Despite the controversy, multiple routes of entry have been tested—PFFs spread efficiently upon intravenous injection [54], injection in the duodenum [122,123] or the olfactory bulb [57], for example, but not upon intravitreal injection [124]. In those cases, the induced pathology mainly mirrors PD. New data from Wang et al. [125] demonstrated that the accumulation of αSyn aggregates in the detrusor and external urethral sphincter nerves arises not in PD patients but in MSA patients exclusively. They also show in a transgenic mice model that PFF injection in the lower urogenital tract led to the development of αSyn inclusions in the brain—a pattern overlapping the αSyn spread observed in MSA [125]. Symptoms, including ataxia and paralysis, were also more consistent with MSA than PD. This suggests that PD and MSA might have different origins; the gut for PD and the urogenital tract for MSA. Furthermore, the evidence could also suggest that synthetic PFFs acquire their strain specificity upon cell propagation and that the variations in peripheral neurons influence which strain is ultimately created [126], as demonstrated in Figure 5.

## 4. Perspective: The Future of Strainotyping

The fact that αSyn aggregates exist as distinctly structured filaments, combined with autonomous biochemical and biophysical characteristics, alongside unique pathological effects, alludes to a tempting hypothesis that could open new doors in the fundamental understanding of synucleinopathies. In turn, the development of methodologies for the rapid and simple assessment of fibril polymorphism, or “strainotyping”, will ameliorate diagnosing capabilities and therapeutic treatments. Currently, strainotyping is primarily conducted by analyzing the biophysical properties of the fibrils via protease resistance testing, examining the cellular/organismal effect of αSyn aggregation, and/or determining the assembly’s structures through electron microscopy. These methods present severe limitations that would preclude their general use in the community, including lack of sensitivity (protease resistance analysis would be difficult to perform on mixed species) or being time-consuming and/or expensive. However, recent studies give clues that easy strainotyping may be achievable in the near future. Structural differences between strains [98,127] can be used to generate conformational antibodies [110,128] to achieve a straightforward recognition of subtypes in the future. This would require existing and newly created antibodies to be thoroughly tested against well-defined polymorphs to avoid cross-reactivity [129]. Multiple groups have also shown that the kinetics of in vitro amplification varied significantly when MSA, DLB or PD-derived αSyn preparations were used [23,40,130,131,132]. Further, the amplified fibrils had different affinities for specific dyes [58,133,134], which would also create a unique signature for each strain.

### 4.1. Reproducibility and Quality Control

Understanding the cellular aggregation, cell-to-cell propagation, or the overall progression of αSyn pathology relies strongly on observing the effects of adding/injecting synthetic αSyn fibrils [135]. As amply described, it is well established that αSyn strains or polymorphs may induce dramatically different effects on cells and tissues [136]. Therefore, it is plausible that disparities between reports arise partially from slight variations in the preparation and handling of the αSyn PFFs. To ensure the reproducibility of biological data, the first step would be to ensure that the PFFs are adequately controlled. The field has long advocated for the necessity to establish unanimous standards regarding both the biological models and reagents, and there is no reason strainotyping would not be an intrinsic addition to said criteria.

### 4.2. Diagnosis and Patients Stratification

Interestingly, the seed amplification assays also react differently to the various syneucleinopathies [23]. DLB samples provide the fastest response [131], while MSA samples, in general, reach a lower signal at saturation [40]. Soto’s group [40] has recently demonstrated that their assay could successfully discriminate between PD and MSA with high sensitivity (95.4%), leveraging differences in kinetics and dye reactivity. Furthermore, it was recently studied that PD-, MSA-, and DLB-positive brain homogenates derived from patient samples underwent SAA and revealed disease-specific differences in the shape of the fibrils [50]. From TEM and Proteinase K SDS-PAGE analysis, it was found that PD and MSA patient-derived αSyn fibrils obtained by SAA exhibited a relatively flat and twisted appearance, significantly resembling the fibrillar polymorph “Ribbons” compared to those derived from DLB patients, which exhibited no twists, were cylindrical, and otherwise indistinguishable from the fibrillar polymorph “Fibrils”. PD and MSA αSyn fibrils exhibited similar digestion profiles that also differed from DLB patients [50]. Notably, the DLB group also demonstrated the most intracellular inclusions, followed by the MSA patients, and then the PD group when each respective αSyn strain was exposed to H4 cells quantified by the number of YFP^+^ puncta per H4 cell nuclei [50]. As strainotyping expands, one can expect distinct synucleinopathy diagnoses to become more common. Given the fact that synucleinopathies can be viewed as a wide spectrum of diseases rather than well-defined pathologies, strainotyping could also provide a refined molecular diagnosis.

### 4.3. Drug Development

From a therapeutic aspect, it is established that not all synucleinopathies respond to the same therapies. For example, MSA is known to be far less responsive, if at all, to L-Dopa treatment compared to PD [1]. Currently, the main axes to treat PD, and by extension synucleinopathies, by preventing αSyn aggregation, inhibiting cell-to-cell propagation, and/or promoting the degradation of αSyn aggregates [137]. These targeted aspects are indeed highly variable between strains, and as therapies evolve from treating the symptoms to addressing the root causes of each disease, it is expected that these biochemical strain variations will strongly impact both the response and efficiency of prospective therapeutic drugs. One hurdle future therapeutics will have to circumvent is the idea that aggregation inhibitors will be impacted by the difference in kinetics observed [23], with rapid nucleation or elongation increasing the risk of therapeutic escape. Another is the use of different entry receptors and overall conformational modifications that would affect the efficiency of both small molecules (targeting the receptor), and antibodies (targeting the fibril), designed to interfere with cellular propagation. Finally, some strains may facilitate degradation avoidance if molecular recognition by the actors of the pathway is affected [56,99]. Overall, this points to the need to test future drugs on different strain models early in the drug development pathway to anticipate the therapeutic effect and select patients. Again, this goes hand in hand with the ability to provide a precise molecular diagnosis, not only between synucleinopathies but within pathologies as well.

## 5. Conclusions

New evidence strongly supports the existence of αSyn strains at the core of the varying synucleinopathies. Strain-specific pathologies could provide a unifying theory and explain the observed phenotypic disparities between diseases, all while recognizing the predominant role of the intrinsically disordered, self-organizing αSyn protein. This perspective reinforces the need for the definitions and standards that will ultimately improve the fundamental understanding of αSyn propagation at the cellular level. Furthermore, the need for further structural, biochemical, and biophysical characterization of synucleinopathy strains is well established, while effective assays for the rapid strainotyping of aggregates are urgently required to advance our understanding of these complex pathologies. Once established, these assays hold the power to facilitate drug development and discovery while pushing therapeutic advancements forward to ultimately ameliorate the lives of patients with Parkinson’s Disease, multiple system atrophy, or dementia with Lewy bodies.

## Figures and Tables

**Figure 1 ijms-24-12134-f001:**
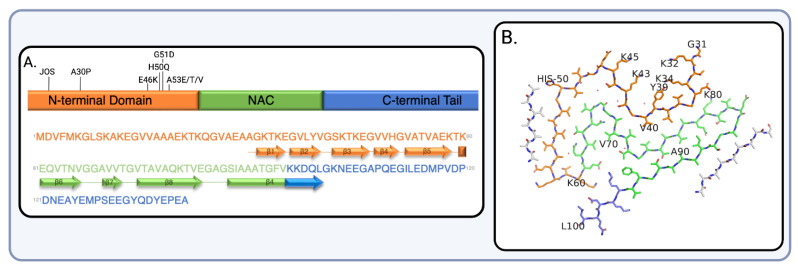
(**A**) Top: Human αSyn amino acid sequence with familial mutations (A30P, E46K, H50Q, G51D, A53E/T/V) [12] and recently discovered juvenile onset synucleinopathy (JOS) insertion [13]. Bottom: The thick connecting lines with arrowheads indicate the presence of β-strands. The N-terminal domain (residues 1–60) is presented in orange, with the NAC region (residues 61–95) in green and the C-terminal region (residues 96–140) in blue. (**B**) The atomic model of αSyn Lewy fold based on Cryo-EM density map. The filament core extends from G31–L100 (PDB 8A9L). Created with BioRender.com (accessed on 25 June 2023).

**Figure 2 ijms-24-12134-f002:**
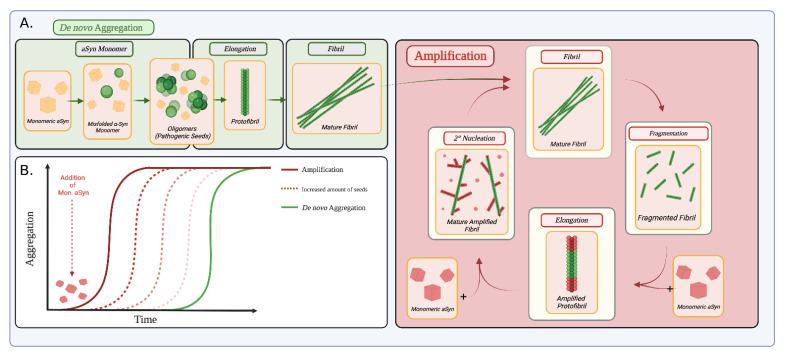
Aggregation of αSyn. (**A**) αSyn aggregates can be amplified by two mechanisms—de novo aggregation in green and amplification in red. Monomeric αSyn can become misfolded and form oligomers that elongate into protofibril and eventually mature fibrils. When fibrils are added to a solution with a new monomer, they can amplify. Fibrils can then be fragmented into seeds which can be elongated by recruiting monomeric αSyn. Secondary nucleation can occur where new fibrils grow at the surface of existing mature fibrils. Amplification is a cyclic process. (**B**) This process is leveraged in seed amplification assays (SAA)—The graph above is a kinetics representation of the fibrillation of αSyn by amplification. With the increased amount of seeds, fibrillation occurs faster. De novo Aggregation (represented in green) also produces mature fibrils, although at a slower overall rate than amplification. Created with BioRender.com (accessed on 25 June 2023).

**Figure 3 ijms-24-12134-f003:**
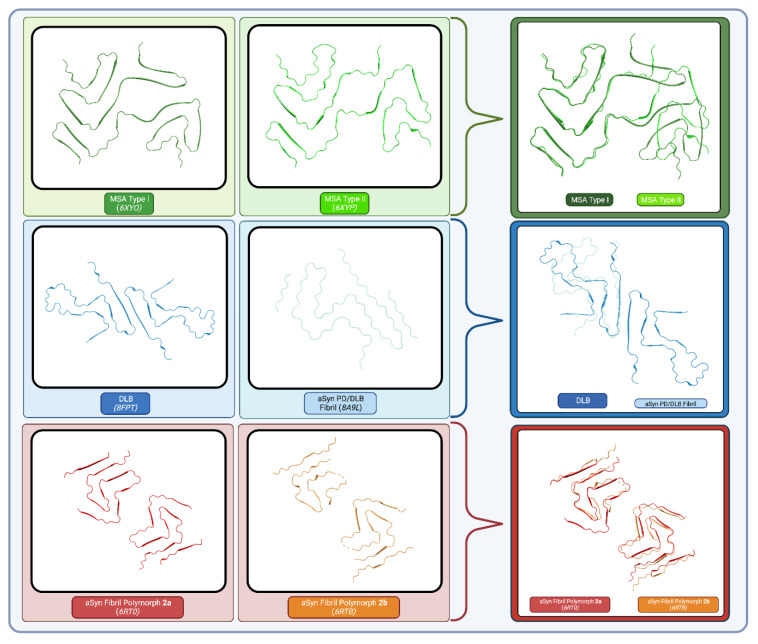
Structure of the fibrils associated with varying synucleinopathies and their respective overlay. MSA Type I (6XYO, 2.60 Å) and MSA Type II (6XYP, 3.29 Å), represented in green, were solved by Cryo-EM. DLB (8FPT) PD/DLB (8A9L, 2.20 Å) represented in blue were solved by NMR and Cryo-EM, respectively, with 8FPT and half of 8A9L overlayed. Synthetic αSyn Fibril polymorph 2a (6RT0, 3.10 Å) and αSyn Fibril Polymorph 2b (6RTB, 3.46 Å) represented in red and orange, respectively were solved by Cryo-EM. Created with BioRender.com (accessed on 25 June 2023).

**Figure 4 ijms-24-12134-f004:**
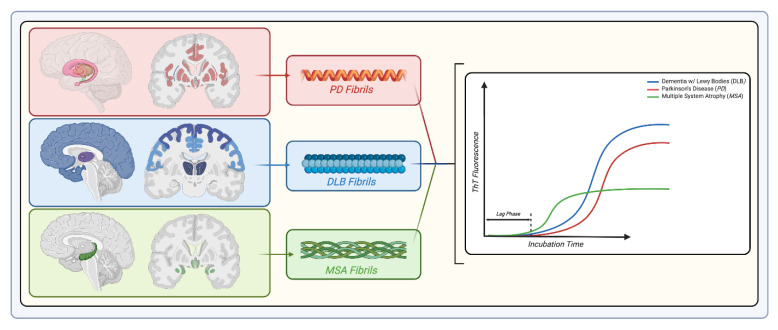
Different synucleinopathies represented in varying brain regions sequestered by contrasting fibril structures. PD, DLB, and MSA are represented in red, blue, and green, respectively. PD is observed in the amygdala, DLB in the cerebral cortex, and MSA in the hippocampus. Each synucleinopathy is associated with a different strain of αSyn fibrils, each with unique biophysical and biochemical properties. In the right-most panel, the varied response to seed amplification assay based on ThT fluorescence is detailed from the differing fibrils associated with each disease. Created with BioRender.com (accessed on 25 June 2023).

**Figure 5 ijms-24-12134-f005:**
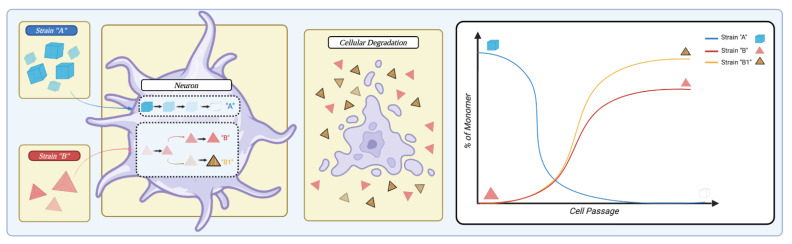
Representation of the cellular ability to influence strain selection. When a cell is in the presence of two different strains (“strain A” in blue and “strain B” in red), it possesses the ability to influence strain propagation. The cell could preferentially degrade strain A while simultaneously amplifying strain B. Further, the cell can modify strain B to generate a new subtype exemplified by strain B1 in striped yellow. Upon cellular degradation, strains are released that can re-infect a different cell. Overall, multiple cell passages can dramatically alter the landscape of strain representation, as shown in the right panel. Created with BioRender.com (accessed on 25 June 2023).

**Table 1 ijms-24-12134-t001:** Summary of αSyn ribbon, fibril, and polymorph aggregates produced under varying conditions across multiple studies. The αSyn Strain, constituent protein, growth conditions and references to each study are included.

αSyn Strain	Protein	Conditions	References
“ribbons”	WT	5 mM Tris-HCl pH 7.5	Peelaerts et al. [54]
“fibrils”	WT	(50 mM Tris-HCl, pH 7.5, 150 mM KCl) at 37 °C under continuous shaking in an Eppendorf Thermomixer set at 600 r.p.m.	Liu et al. [55]
“ribbons”	WT	30 mM Tris-HCl, pH 7.5	Suzuki et al. [56]
“fibrils”	WT	30 mM Tris-HCl, pH 7.5, containing 150 mM KCl and 0.1% NaN_3_, to a final concentration of 6 mg/mL. The samples were incubated at 37 °C under rotation at 20 rpm for 7 days.	Suzuki et al. [56]
F-65	WT	50 mM MES pH 6.5, 150 mM NaCl	Rey et al. [57]
F-91	WT	25 mM Na_2_PO_4_ pH 9.1	Rey et al. [57]
F-110	WT (aa 1–110) C-terminal truncated	40 mM TrisHCl pH 7.5, 150 mM KCl.	Rey et al. [57]
Low Salt “NS”	A53T	20 mM Tris-HCl, pH 7.4	Lau et al. [58]
High Salt “S”	A53T	20 mM Tris-HCl, pH 7.4 and 100 mM NaCl	Lau et al. [58]

## Data Availability

No new data were created or analyzed in this study. Data sharing is not applicable to this article.

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
