# Peer review of "α-Synuclein Strains and Their Relevance to Parkinson’s Disease, Multiple System Atrophy, and Dementia with Lewy Bodies"

_ijms, 2023, doi:10.3390/ijms241512134_

Round 1
Reviewer 1 Report
With their review article "Alpha-synuclein strains and their relevance to Parkinson's disease, Multiple System Atrophy and Dementia with Lewy bodies", the authors provide an excellent overview on the state-of-the-art of alpha-synuclein aggregation. The current knowledge on disease- and inclusion type-specific formation of alpha-synuclein strains are summarized and discussed in the context of alpha-synuclein aggregate amplification and its diagnostic potential.
The different strains of aggregated alpha-synuclein are compared in their aggregation, propagation and neurotoxicity properties including the most recent and relevant papers.
I have only three minor comments.
1. Page 8, line 290: PFFs, the abbreviation is used before its definition in lines 300-301. Please, change it.
2. Page 9, Second paragraph: I would recommend to include a very brief description of the structural differences between alpha-synuclein polymorphs referred to as ribbons and fibrils.
3. Page 16, line 647: Should it be rather than promoting alpha-synuclein degradation, promoting the degradation of alpha-synuclein aggregates?
Reviewer 2 Report
Page 5, line 185 “Amyloid fibrils are a common fold in nature.” – This is incorrect sentence. Amyloid fibrils are highly polymorphic structures. And so far, we cannot say something concrete about these structures. So far, there are only models. Cryo EM also gives some models, since the model structure is included in the calculations.
The model of Infinite beta-sheets is a model, no one can build such a structure in cells, for this you need a special apparatus, like for actin filaments, for the construction of which more than 100 proteins are involved in the cell. It is difficult to imagine that there is a relationship between the type of disease (Parkinson's disease and multiple system atrophy) and fibril morphology when we know that there is huge polymorphism for amyloid fibrils (even for the same sample). The morphology of the fibrils in article 39, figure 3 is the same, how it is possible to obtain a different stacking of monomers in a fibril from this remains only a big mystery. The same applies to figure 7 in article 40, the morphology of the fibrils is very similar, it is very difficult to talk about the structural polymorphism of these fibrils.
The authors wrote that :”Previously, heparan sulphate proteogly-439 cans (HSPGs)78, and particularly syndecan 379, have been shown to bind and mediate the 440 uptake of αSyn aggregates.”
In this case for syndecan 2 it has been suggested that it may mediate Abeta fibril formation (PMID: 25916113 Free article. ).
The reference system is different and there should be no references in the abstract.
Page 2 line 61 “in MSA1,2The 61” Please correct.
Line 94 “interactions8,9.” ” Please correct.
Reviewer 3 Report
α-Synuclein strains and their relevance to Parkinson’s disease, Multiple System Atrophy, and Dementia with Lewy Bodies
Synopsis: the article is about is about the role of Alpha-synuclein (αSyn) aggregates in the onset and progression in a group of diseases defined synucleinopathies. After an introductive part, the pathological aggregation of αSyn is described, highlighting how different forms of synucleinopathies have peculiar and characteristic features.
Critic: the article analyses the phenomenon of synucleinopathies from all possible angles and with thoroughness. Although the form cannot be said to be incorrect in absolute terms, the article alternates between formal and informal expressions when discussing the topic. Several punctuation errors are also present. Although the references used are believed to be valid, in some cases lengthy items were written using only one or two of them. The journal's criteria for references are not met.
Major
According to editorial rules, In the text, reference numbers should be placed in square brackets [ ]. Also, bibliography is incorrect.
In captions, the date of use of biorender tool is not indicated, nor its URL.
Minor
Several punctuation errors are present.
αSyn requires to be written in extended form the first time is used in the text body.
Parkinson’s disease is not always indicated with its acronym.
In some cases, αSyn is indicated as just Syn.
In some cases, Lewy body and Lewy neurite are written in extended form.
Line 65: one or more references are required.
Line 79: one or more references are required.
Line 88: “αSyn is a small 140 amino-acid protein, expressed throughout the brain and peripheral nerves as well as in other tissues at lower levels”. This is not totally correct, being it highly express in bone marrow and lymphoid tissue. A reference is also needed.
Line 183: one or more references are required.
Line 257: one or more references are required.
Line 296: one or more references are required.
Figure 1 caption: references are not in order. Number 128 and 129 should be the 18 and 19, respectively.
Minor editing of English language required
Reviewer 4 Report
A fundamental issue that is addressed in this review article is the "urgent need of developing assays for strainotyping of aggregates". Therefore, it should be reflected and included in the abstract (at the end). I suggest a short sentence so as not to increase too much the maximum number of characters imposed by the journal editorial on the length of the abstract.
«Additionally, we will highlight the need of effective methodologies for strainotyping of aggregates to ameliorate diagnosing capabilities and therapeutic treatments.»
Reviewer 5 Report
In their paper entitled “alpha-Synuclein strains and their relevance to Parkinson’s disease, Multiple System Atrophy, and Dementia with Lewy Bodies”, the Authors report on a comparison among Parkinson Disease (PD), Dementia with Lewy Bodies (DLB), and Multiple System Atrophy (MSA), in relation with the formation, and consequently with the involvement, of aSyn aggregates in each of these neurodegenerative disorders.
The paper is of interest and suitable for International Journal of Molecular Sciences. The state of art is well described, and many interesting and recent findings have been reported. Bibliography is up-to-date: 67 (out of 129) of the cited papers have been, indeed, published between 2020 and 2023.
Of particular interest is the suggestion that different alpha-syn strains are produced in different pathologies, and that this strains somehow “imprint” the pathology, and can transmit it in a specific manner, with differences even depending on the brain origin of the cell type treated.
I only have a brief suggestion for the Authors: many reports have shown that, as in the case of prions, extracellular vesicles (EVs) are involved in the cell-to-cell transfer of alpha-syn aggregates; it could be of interest to add a few sentences on this point.
Round 2
Reviewer 2 Report
In my previous remark: "The authors wrote that :”Previously, heparan sulphate proteogly-439 cans (HSPGs)78, and particularly syndecan 379, have been shown to bind and mediate the 440 uptake of αSyn aggregates.”
In this case for syndecan 2 it has been suggested that it may mediate Abeta fibril formation (PMID: 25916113 Free article.)."
the authors have added the sentence (above). But I propose to add the mentioned reference PMID: 25916113, where the authors speculated about the role of syndecan 2 in 2015, but the authors of 85 reference demonstrated this only in 2019.
We added “Syndecans have also been suggested to play a role in mediating fibril formation of not only αSyn, but also Tau[84] and, Ab fibrils[85].”(page 12, lines 451-453)
Author Response
We have added reference (E. I. Leonova and O. V. Galzitskaya, ‘The role of syndecan-2 in amyloid plaque formation’,) to the following sentence: “Syndecans have also been suggested to play a role in mediating fibril formation of not only αSyn [85], …” (Page 11, line 452).
Reviewer 3 Report
Reviewer 3 Comments:
Synopsis: the article is about is about the role of Alpha-synuclein (αSyn) aggregates in the onset and progression in a group of diseases defined synucleinopathies. After an introductive part, the pathological aggregation of αSyn is described, highlighting how different forms of synucleinopathies have peculiar and characteristic features.
Critic: the article analyses the phenomenon of synucleinopathies from all possible angles and with thoroughness. Although the form cannot be said to be incorrect in absolute terms, the article alternates between formal and informal expressions when discussing the topic. Several punctuation errors are also present. Although the references used are believed to be valid, in some cases lengthy items were written using only one or two of them. The journal's criteria for references are not met.
Second revision: the article has been greatly improved but some problems about the way the references are reported still exist. A space is required before each parenthesis, and if there are several references, they must all be placed together within the same parenthesis. Once again, the authors do not comply with the editorial criteria of IJMS in the bibliography.
Major
According to editorial rules, In the text, reference numbers should be placed in square brackets [ ]. Also, bibliography is incorrect.
Updated citation style from “Nature” to “IEEE” altering in-text citations to “[x]” format. Bibliography has been updated.
NOT DONE
In captions, the date of use of biorender tool is not indicated, nor its URL.
Because each figure has been created as an original figure without the use of BioRender templates, as per BioRender, we are only needed to include “Created with BioRender.com” in the citation as is already included. Please view https://help.biorender.com/en/articles/3619405-how-do-i-cite-biorender for more information.
DONE
Minor
Several punctuation errors are present.
Corrected.
DONE
αSyn requires to be written in extended form the first time is used in the text body.
Although “α-synuclein” is written in the abstract (page 1, line 12), we added the extended form of αSyn in the introduction on page 2, line 39.
DONE
Parkinson’s disease is not always indicated with its acronym.
Corrected, page 4 line 133, 134, 149, 153, 155.
DONE
In some cases, αSyn is indicated as just Syn.
Corrected, page 7, line 236, page 7, line 244, page 8, line 294, page 9, line 324, page 9, line 353, page 11, line 426,
DONE
In some cases, Lewy body and Lewy neurite are written in extended form.
Lewy Bodies and Lewy Neurites first mentioned on page 2, line 76. The next time they are mentioned in their extended form is page 9, line 338 where they are conjunctively listed together as “Lewy Body/Lewy Neurite-like inclusions”. Has been adjusted to “LB/LN-like inclusions…”
DONE
Line 65: one or more references are required.
Changed “can induce psychosis” to “may induce psychosis” and added reference [5] (Zahodne, Fernandez, 2010) page 2, line 66.
DONE
Line 79: one or more references are required.
Added reference [7] (Malfertheiner et. Al., 2021) to page 2, line 79
DONE
Line 88: “αSyn is a small 140 amino-acid protein, expressed throughout the brain and peripheral nerves as well as in other tissues at lower levels”. This is not totally correct, being it highly express in bone marrow and lymphoid tissue. A reference is also needed.
Corrected “lower levels” to “varying levels” page 2, line 90. Added reference [9] (Yang et al. 2022) page 2, line 90.
DONE
Line 183: one or more references are required.
Added [32] (Jung et Al., 2017) to page 5, lines 185 and 187
DONE
Line 257: one or more references are required.
Added [17] (Ma et. al., 2019) to page 7, line 260 and [44] Prusiner et al., 2015) to page 7, line 262
DONE
Line 296: one or more references are required.
Added [54] (Peelaerts et al., 2015] to Page 8, line 302
DONE
Figure 1 caption: references are not in order. Number 128 and 129 should be the 18 and 19, respectively.
Added “Additionally, the N-terminal region has been shown to possess familial mutations [12] and more recently established, a juvenile onset synucleinopathy insertion in the same region [13].” on page 2-3, lines 97-99 in text thus updating figure captions accordingly.
DONE
Minor editing of English language required
Author Response
A space has been added to every in-text citation. Citation style updated to “American Chemical Society (ACS)” as per MDPI citation guide with modifications on inline citations (“xx” to “[x]”). Also removed “et al.” from after 2 authors to after 10 and includes multiple in text citations within the same brackets as per MDPI standards. For citations that are “x” and “x”, a comma (,) (e.g. [1,2]) is used whereas citations that are “x” to “x”, a dash (-) is used (e.g. [3-5], includes 3,4, and 5) and placed inside the punctuation as per the MDPI citation guide.
We updated bibliography abbreviations of journals as per MDPI requirements (“Journal of Biological Chemistry” to J. Biol. Chem, [47], [87], [98],[102], “Journal of Neuroscience” to “J. Neuro. [35],[59],[103],[109], “Proceedings of the National Academy of Sciences” to “PNAS” [67],[78], “The Lancet” to “Lancet” [2],[25],[71],[73], “The international journal of biochemistry and cell biology” to “Int. J. Biochem” [76], “Movement Disorders” to “J. Mov. Disord” [24],[31],[119]).